# Development of a Dual-Readout Multicolor Immunoassay for the Rapid Analysis of Isocarbophos in Vegetable and Fruit Samples

**DOI:** 10.3390/foods13244057

**Published:** 2024-12-16

**Authors:** Zijian Chen, Wei-Xuan Huang, Hongwu Wang, Meiling Zhang, Kai Chen, Hao Deng

**Affiliations:** 1Key Laboratory of Tropical Fruit and Vegetable Cold-Chain of Hainan Province, Institute of Agro-Products of Processing and Design, Hainan Academy of Agricultural Sciences, Haikou 571100, China; 2School of Food & Pharmaceutical Engineering, Zhaoqing University, Zhaoqing 526061, China; chenzijian@zqu.edu.cn (Z.C.); hwwang@zqu.edu.cn (H.W.); zml13430117492@163.com (M.Z.); blacoolker@163.com (K.C.); 3Laboratory of Quality & Safety Risk Assessment for Agro-Products (Zhaoqing), Ministry of Agriculture and Rural Affairs, Zhaoqing 526061, China; 4Guangdong Engineering Technology Research Center of Food & Agricultural Product Safety Analysis and Testing, Zhaoqing 526061, China; 5Department of Electrical Engineering, City University of Hong Kong, Hong Kong 999077, China; weixhuang6-c@my.cityu.edu.hk

**Keywords:** pesticide, multicolor immunoassay, isocarbophos, alkaline phosphase, RGB analysis

## Abstract

Multicolor immunoassay is a powerful tool for rapid analysis without the use of bulky instruments owing to various color conversions, which is suitable for on-site visual analysis for pesticides. Herein, this study developed a multicolor immunoassay for the rapid detection of isocarbophos. After competitive immunoassay, the secondary antibody (GAM-ALP) catalyzed ascorbyl-2-phosphate (AAP) into ascorbic acid (AA). The AA can reduce K_3_[Fe(CN)_6_] into K_4_[Fe(CN)_6_]. The latter can react with Fe^3+^ to form Prussian blue; otherwise, the orange AAP-Fe^3+^ complex was generated. Therefore, the multicolor immunoassay achieved a color conversion of orange–green–blue in response to isocarbophos, allowing for rapid semiquantitative analysis by the naked eye. After parameter optimization, the multicolor immunoassay was developed depending on the ratiometric absorbance between the Prussian blue and AAP-Fe^3+^ complex. Moreover, a smartphone was used to measure the RGB value of the color conversion for the development of portable visual, quantitative analysis. Both the absorbance-based and RGB-based multicolor immunoassays showed good accuracy and practicability in the recovery test. This study provided a common approach for the development of dual-readout multicolor immunoassay, which can be used for on-site rapid screening by quantitative or visual semiquantitative analysis.

## 1. Introduction

Pesticides are an indispensable element for modern agriculture to ensure crop yield [1,2]. However, the inappropriate and excessive application of pesticides can lead to the persistence of their residues not only in agricultural produce but also within the soil and water, which poses a potential risk of pesticide exposure and human health with a range of adverse effects [3,4]. Organophosphorus pesticides (OPs) are extensively employed in agricultural practices to control agricultural pest insects and are recognized for their remarkable efficiency and extensive applicability [5,6]. Isocarbophos is one of the most used highly toxic OPs in rice fields to control chewing and sucking insects, which, however, cause liver dysfunction, vascular dementia, cancer, and other toxicity [7,8,9]. Owing to the hazard to human beings, the use of isocarbophos has been prohibited by numerous countries and organizations, such as China, the USA, and the European Union. The instrumental method (such as HPLC-MS/MS or GC-MS/MS) is the conversional analysis method for isocarbophos. However, agricultural production is decentralized, and samples must be collected and sent back to central laboratories for instrumental analysis, which are laborious, time-consuming, and high-cost. Alternatively, immunoassay or immunosensor is a powerful tool for the screening of harmful organic molecules with the advantage of being rapid, straightforward, and cost-effective, and it has been widely employed for pesticide analysis [10,11,12].

Conventional immunoassay, including enzyme-linked immunosorbent assay (ELISA) and immunochromatography analysis (ICA), depends on the signal color signal changes from deep to light or light to deep [13]. Nevertheless, human eyes are more sensitive to variations in color than in intensity density [14]. Recently, colorimetric immunoassay has emerged as a promising approach, achieving semiquantitative visual analysis by the naked eye, thereby eliminating the requirement for bulky equipment [15,16]. Precious metal-based (such as gold or silver) multicolor immunoassay has been widely reported [16,17]. However, the high cost of reagent, laborious absorption peak scanning, and low absorbance caused by blue-shift limit the precious metal-based multicolor immunoassay [18,19,20]. Alternatively, a multicolor immunosensor relying on the direct color transition stemming from the ratiometric variation between two fixed wavelength signals, be it absorbance or fluorescence [21,22,23], offers a more streamlined and time-efficient approach compared to previously reported precious metal-based multicolor immunosensors. Moreover, by incorporating smartphone-based and RGB color space recognition, multicolor immunoassays can achieve precise quantification without bulky equipment [24,25,26].

In a previous study, an anti-isocarbophos monoclonal antibody (mAb) was prepared in our laboratory [27]. Herein, the anti-isocarbophos mAb would be employed to develop a multicolor immunoassay. The goat-anti-mouse alkaline phosphatase-conjugated (GAM-ALP) and ascorbyl-2-phosphate (AAP) were used as the signal probe and substrate, respectively. The K_3_[Fe(CN)_6_] and Fe^3+^ were used as the color-developing agent for the development of multicolor immunoassay. Moreover, we also developed software to rapidly measure the RGB value in the well of the microplate array for quantitative analysis. The practicality and accuracy of the multicolor immunoassay were further accessed by standard HPLC-MS/MS.

## 2. Materials and Methods

### 2.1. Materials

Isocarbophos standard was obtained from Tanmo Standard Center (Beijing, China). The ascorbic acid (AA), ascorbyl-2-phosphate (AAP), Tris (hydroxymethyl) aminomethane (Tris), MgCl_2_, K_3_[Fe(CN)_6_], FeCl_3_ were purchased from Aladdin Co., Ltd. (Shanghai, China). Alkaline phosphatase was supplied by Yuanye Bio-Technology Co., Ltd. (Shanghai, China). Acetic acid and Tween-20 were supplied from Damao Chemical Technology Co., Ltd. (Tianjin, China). The secondary antibody goat-anti-mouse alkaline phosphatase-conjugated (GAM-ALP) was purchased from Biosharp Co., Ltd. (Hefei, China). The anti-isocarbophos monoclonal antibody (mAb) was prepared from our previous study.

### 2.2. Instruments

Absorbance was measured using an HBS-Scan Y microplate reader (DeTie Biotechnology Co., Ltd., Nanjing, China). The 96-well microplates were washed in an HBS-4009 microplate washer (DeTie Biotechnology Co., Ltd., China). Centrifugation (TGL-15B) was purchased from Anting Scientific Instrument Factory (Shanghai, China). HPLC-MS/MS (1290 Infinity-6495) was performed by Agilent Technologies, Inc. (Santa Clara, CA, USA) using a liquid chromatography column (Welch, 100 × 2.1 mm, 3.5 μm). The morphology of the generated Prussian blue was analyzed by transmission electron microscope (TEM) (FEI-TALOS-F200X, Thermo Fisher Scientific, Waltham, MA, USA) and scanning electron microscope (SEM) (SU8010, Hitachi, Tokyo, Japan). The hydrated particle size of Prussian blue was measured by dynamic light scattering (DLS) (Nano ZS90, Malvern, UK). The measurement of the Fourier transform infrared spectroscopy (FTIR) spectrum of Prussian blue was performed by FTIR-650s (Gangdong Sci. & Tech. Co., Ltd., Shantou, China). The X-ray diffraction (XRD) and the X-ray photoelectron spectroscopy (XPS) of Prussian blue were analyzed by XRD-6100 (Shimadzu, Kyoto, Japan) and K-Alpha X (Thermo Fisher Scientific, USA).

### 2.3. Buffers

The following buffers and solutions were used in this study: (1) phosphate-buffered saline (PBS, 10 mM, pH 7.4, with 50 mM NaCl) was used for anti-isocarbophos monoclonal antibody dilution; (2) Tris buffer (10 mM, pH 7.4, with 0.05% Tween-20) was used for GAM-ALP dilution and washing microplate; (3) ALP buffer (5 mM Tris, pH 9.5, with 2 mM MgCl_2_) was used for the catalysis of AAP; (4) acetate buffer (100 mM, pH 4.0) was used to adjust pH for generation of Prussian blue.

### 2.4. Development of Multicolor Immunoassay

#### 2.4.1. Absorbance-Based Mode

A series concentration of isocarbophos or sample extraction was added as a competitor to the antigen-coated 96-well microplate (50 μL/well), and the diluted mAb was subsequently added (50 μL/well) for 40 min incubation at 37 °C. Afterward, the plate was washed five times with washing buffer, and GAM-ALP conjugation was added (100 μL/well) for 30 min incubation at 37 °C. After five times washing, the AAP (5 mM, in ALP buffer) was added (100 μL/well) for 40 min incubation at 37 °C. Afterward, K_3_[Fe(CN)_6_] (2 mM, in acetate buffer) was added (50 μL/well) for a 10 min reaction. Afterward, FeCl_3_ (2 mM, in acetate buffer) was added (50 μL/well) for another 10 min reaction. Then, the absorbance at 715 nm (A_715_) and 425 nm (A_425_) were measured, respectively. The calibration curve was fitted using OriginPro2024 software with the value of A_715_/A_425_ against the concentration of isocarbophos.

#### 2.4.2. RGB-Based Mode

The 96-well microplate was placed on the luminous plate and then photographed by smartphone. The script then batch processes all images within the input folder. Each image was cropped to remove unnecessary outer areas while preserving the central board and part of the white background. This ensures consistency in the region of interest across different images, eliminating background noise and focusing on the target area for accurate color extraction.

The two modes of multicolor immunoassay required K_3_[Fe(CN)_6_] and FeCl_3_ as color-developing agents instead of precious metals to generate color conversion, which achieved lower cost and faster detection time.

### 2.5. Recovery Test

#### 2.5.1. Sample Pretreatment

Three levels of isocarbophos (0.1, 0.3, and 1.0 mg/kg) were added to 10 g of samples (cucumber, lettuce, and orange), respectively. Samples were homogenized and mixed with 10 mL of acetonitrile in a 50 mL polypropylene centrifuge tube. Four grams of MgSO_4_, 1 g of NaCl, 1 g of sodium citrate, 0.5 g of sodium dihydrogen citrate, and a ceramic homogenizer were added, and the tube was shaken vigorously for 1 min, followed by centrifuging for 5 min at 1676× *g*. Six milliliters of the upper layer were transferred to a 15 mL polypropylene centrifuge tube with the addition of 900 mg of MgSO_4_ and 150 mg of PSA for 1 min vigorously shaken, followed by centrifuging for 5 min at 1676× *g*. The extractions were filtrated by 0.22 μm membrane for HPLC-MS/MS analysis and 10-fold diluted for multicolor immunoassay.

#### 2.5.2. Parameters of HPLC

Phases A: 0.1261 g of ammonium formate was dissolved in 0.01% formic acid aqueous solution and diluted to 1000 mL. Phases B: 0.1261 g of ammonium formate was dissolved in 0.01% formic acid methanol solution and diluted to 1000 mL. The HPLC-MS/MS was applied by Agilent 1290 Infinity-6495 (ESI source), and the liquid chromatographic column was Welch (100 × 2.1 mm, 3.5 μm). The gradient elution parameters are summarized in Appendix A. The flow rate, column temperature, and injection volume in HPLC were 0.3 mL/min, 40 °C, and 2 μL, respectively.

#### 2.5.3. Parameters of Mass Spectrometry

The type of ion source was electrospray ionization (ESI), and the polarity for all the pesticides was positive. The acquisition mode for mass was multiple reaction monitoring (MRM). The ion pair for mass analysis for isocarbophos are 231/121 and 231/109.

## 3. Results and Discussion

### 3.1. Feasibility of Multicolor Immunoassay

The strategy of multicolor immunoassay is shown in Figure 1A. After indirect competitive immunoassay, the binding GAM-ALP in microplate catalyzed substrate AAP into ascorbic acid (AA), the later can reduce K_3_[Fe(CN)_6_] into K_4_[Fe(CN)_6_] that reacted with Fe^3+^ to form Prussian blue through self-assembly. While in the presence of isocarbophos, mAb was inhibited, and thereby, no Prussian blue was formed. The AAP can chelate Fe^3+^ to form the orange AAP-Fe^3+^ complex. Therefore, an obvious color conversion immunoassay from blue to orange that is related to the concentration of isocarbophos would be developed.

To investigate the feasibility of multicolor immunoassay, the absorbance of color reaction for related reagents was analyzed. As shown in Figure 1B,C, the Fe^3+^ showed no obvious absorbance in the range from 450 nm to 900 nm, while K_3_[Fe(CN)_6_] exhibited a weak absorption peak at 420 nm. When Fe^3+^ was mixed with K_3_[Fe(CN)_6_], a broad peak from 400 nm to 425 nm was observed, which contributed to the characteristic absorption spectra of Fe^3+^ and K_3_[Fe(CN)_6_]. The mixture of Fe^3+^ and AAP showed an obvious orange color and the absorption peak at 460 nm. In the presence of K_3_[Fe(CN)_6_], the intensity of the absorption peak improved, and the peak shifted to 425 nm (A_425_) while the orange color was still maintained. The clear solution was observed in well 6 (Figure 1B,C). The ALP was premixed with substrate AAP to produce AA. The latter can reduce K_3_[Fe(CN)_6_] into K_4_[Fe(CN)_6_]. Moreover, the absorption spectrum of well 6 showed no absorption peaks in the range from 400 nm to 900 nm, indicating the complete reduction for K_3_[Fe(CN)_6_]. Well 7 in Figure 1B,C suggested that the produced K_3_[Fe(CN)_6_] reacted with Fe^3+^ to form Prussian blue via self-assembly and exhibited an absorption peak at 715 nm (A_715_). The blue color in well 8 in Figure 1B,C indicated that the ALP-AAP catalytic system can successfully produce Prussian blue. Well 5 and well 8 indicated that a color conversion from orange to blue was induced in the absence of ALP, demonstrating the feasibility of the development of a multicolor immunoassay.

The color conversion response to ALP was investigated. As shown in Figure 1D, with the increasing ALP concentration, the A_715_ increases with the decrease in A_425_. Therefore, the ratio of A_715_/A_425_ can be used for the development of multicolor immunoassay owing to the self-calibration and possessing a higher detection accuracy toward the targets [28]. Figure 1E showed that the solution exhibited blue color when the A_715_/A_245_ ratio was higher than approximately 3, which was employed to define titer.

### 3.2. Characterization of Prussian Blue

#### 3.2.1. Morphology Characterization

The morphology of generated Prussian blue was characterized by SEM. As shown in Figure 2A, the generated Prussian blue mainly formed the agglomeration morphology and few cubic morphologies, which showed good agreement with the results of TEM analysis (Figure 2B). The main reason for the agglomeration morphology might be that the Fe^3+^ and K_4_[Fe(CN)_6_] can only be mixed briefly in the microplate well without vigorous stirring or any surfactant. Consequently, only a few dispersive cubic Prussian blue nanoparticles were observed. Although Prussian blue exhibited irregular agglomeration morphology, the generated Prussian blue showed good uniformity hydrated particle size (approximately 65 nm), which was analyzed by DLS (Figure 2C). The TEM characterization showed the single cubic morphology of the Prussian blue nanoparticle (Figure 2C). Moreover, a lattice fringe with a *d* spacing of 0.36 nm was observed using high-resolution TEM, which was assigned to the (220) plane of Prussian blue [29] (Figure 2D).

#### 3.2.2. Component Characterization

Since the TEM and SEM analysis shows the indistinct cubic morphology of Prussian blue, several structural characterizations, including FTIR, XRD, and XPS, were used to confirm the generation of Prussian blue.

The generated Prussian blue from the multicolor immunoassay was dried and was further analyzed by FTIR, and the results are shown in Figure 2E. The key structure of Prussian blue is the coordinate bond between the cyan group and the ferric or ferrous atom. The obvious peak at 2082 cm^−1^ belongs to the absorption of the cyan group, and the 498cm^−1^ peak demonstrated the structure of Fe^2+^-CN-Fe^3+^ [30,31,32,33]. The broad peak of 3415 cm^−1^ can be allocated to the stretching vibration of hydroxyl group (-OH) [31]. However, no hydroxyl group exists in Prussian blue. The hydroxyl group might belong to ascorbic acid that is produced from AAP. Moreover, the strong peak at 1612 cm^−1^ was performed by the bending vibration of H-O-H, suggesting the existence of interstitial water within Prussian blue [31]. The peak at 1673 cm^−1^ was almost overlapped by the 1612 cm^−1^ peak. The imperceptible peak (1673 cm^−1^) can be allocated to the C=O group from dehydroascorbic acid, which is generated after a reduction reaction between ascorbic acid and potassium ferricyanide.

Afterward, the generated Prussian blue was further analyzed by XRD. As shown in Figure 2F, the diffraction pattern for the Prussian blue has three obvious peaks at 2 theta values of 16.6°, 23.8°, and 34.5°, corresponding to (200), (220), and (400) of Prussian blue, respectively [31]. While weak peaks at 2 theta values of 38.6°, 42.7°, 49.6°, 53.1°, and 56.2°, corresponding to (420), (422), (440), (600), and (620) of Prussian blue, respectively [29,31]. Although the intensity of the diffraction pattern at higher degrees was weak, the (200), (220), (400), and (420) lattice planes of a cubic Prussian blue structure [34] indicated the successful generation of Prussian blue.

The XPS was employed to characterize the element state and structure of the generated Prussian blue. The survey XPS spectra of Prussian blue showed the presence of Fe 2*p*, N 1*s,* and C 1*s* from the generated Prussian blue, and the O 1*s* might be from the absorbed AAP, ascorbic acid, or dehydroascorbic acid (Figure 3A). The elements that constitute Prussian blue are C, N, and Fe, respectively, which were analyzed through high-resolution XPS. The C 1*s* high-resolution spectra exhibited four contributions (Figure 3B). The binding energy (BE) peaks at 284.9 eV, 285.7 eV, and 286.6 eV related to C-C, C-N or C=N, and C=O group, respectively [35,36]. The C-C and C=O groups did not belong to Prussian blue but might instead be absorbed AAP, ascorbic acid, or dehydroascorbic acid. The BE peak of 288.2 eV of C 1*s* is attributed to the existence of -C≡N [37], which is the feature structure of Prussian blue. Figure 2C shows that the N 1 *s* is decomposed into two peaks at the binding energy of 396.98 eV, 398.73 eV, and 401.5 eV, corresponding to Fe-C≡N, Fe-N, and oxidized N, respectively [38]. As shown in Figure 2D, the Fe 2*p* spectrum exhibits two contributions, 2*p*_1/2_ and 2*p*_3/2_. The peaks of Fe^2+^ were observed at 708.62 eV (2*p*_3/2_) and 721.5 eV (2*p*_1/2_), with the satellite peaks at 710 eV (2*p*_3/2_) and 722.8 eV (2*p*_1/2_), respectively. The peaks at 712.8 eV (2*p*_3/2_) and 727.1 eV (2*p*_1/2_) were attributed to the Fe^3+^, whereas the peaks at 715.5 eV and 729 eV were assigned to the satellite peaks of Fe^3+^. All the peaks of the Fe element were assigned to Prussian blue [29,37,38,39].

All the component characterization indicated that the Prussian blue was successfully generated in the ALP-drive color reaction, exhibiting a notable appreciable color response for multicolor immunoassay.

### 3.3. Optimization and Development of Multicolor Immunoassay

Key parameters were optimized before multicolor immunoassay development. The GAM-ALP is the critical signal probe for color conversion, and the dilution ratio was optimized. Figure 4A shows that the A_715_/A_425_ value decreased as the GAM-ALP dilution ratio increased. The highest A_715_/A_425_ value was observed with the GAM-ALP dilution ratio of 1000, which was chosen as the optimized condition. Based on the optimized dilution ratio, AAP concentration was subsequently optimized. As shown in Figure 4B, the A_715_/A_425_ value increased as AAP concentration decreased and reached the highest value in AAP concentration of 5 mM, then subsequently decreased. The main reason for this is that the high concentration of AAP substrate resulted in excess AAP, and thereby, the AAP-Fe^3+^ complex was mainly formed. As AAP decreased, the percentage of AA increased after catalysis, and thereby, more Prussian blue was formed. However, the low concentration of AAP led to a low concentration of AA, resulting in a decrease in the A_715_/A_425_ ratio. Based on the optimized GAM-ALP and AAP concentration, the coating antigen concentration was subsequently optimized. The mAb showed obvious titers with the coating antigen concentration of 2 μg/mL, 1 μg/mL, and 0.5 μg/mL, while no titer was observed for that of 0.25 μg/mL (Figure 4C). The calibration curves for the three concentrations of coating antigen were further investigated. Figure 4D shows that the IC_50_ of the three curves was 23.7 ng/mL, 24.3 ng/mL, and 35.1 ng/mL, respectively. Although curve 1 and curve 2 showed slight differences, curve 1 achieved a higher titer than curve 2. Consequently, the optimized coating antigen concentration was 2 μg/mL.

The calibration curve for multicolor immunoassay was finally created based on the optimized parameters. As shown in Figure 5A, the developed calibration curve exhibited a color conversion from orange to green and then to blue as isocarbophos concentration decreased. The calibration curve showed IC_50_ of 11.5 ng/mL, with the limit of detection (LOD) of 4.6 ng/mL. The linear range for the calibration curve was 3.9 ng/mL to 62.5 ng/mL, which showed good linearity (Figure 5B).

### 3.4. RGB Value Analysis for Multicolor Immunoassay

Since the multicolor immunoassay exhibits apparent multicolor conversion, the RGB value of each well can be measured by smartphone, which can be used for on-site point-of-care analysis without bulky readers [40,41,42]. By eliminating the need for bulky and expensive equipment, this approach makes advanced analysis capabilities accessible even in resource-limited settings. The portability and ease of use of smartphones enable real-time measurements at the field, market, or other relevant testing sites, offering higher analysis efficiency and lower cost.

As shown in Figure 6A, the microplate was placed on the luminous panel before being photographed. The image was analyzed by “color extraction” software to measure the RGB value of the center point of each well. To verify the robustness of the RGB analysis for multicolor immunoassay, the well added with isocarbophos standard was photographed with different shutter speeds, and the RGB value was analyzed. As shown in Figure 6B,C, the B/G mode exhibited high anti-luminance interference capacity, while the B/R mode showed poor stability. Therefore, the B/G mode was used to develop RGB value analysis for multicolor immunoassay. The calibration curve of RGB analysis showed IC_50_ of 31.8 ng/mL with the LOD of 10.9 ng/mL (Figure 7A), and the linear range was 7.8 ng/mL to 125 ng/mL (Figure 7B), which is broader than that of microplate reader mode. Although the RGB analysis showed lower sensitivity than that of reader mode, it can achieve on-site real-time detection. Moreover, the broader linear range further improves the detection efficiency. Therefore, the RGB analysis exhibited the advantages of simplicity, convenience, and low cost, which was the alternative solution for rapid quantitative analysis. And the detection performance of this multicolor immunoassay was superior to comparable with most reported immunoassay methods for isocarbophos analysis (Appendix A). The sensitivity of the developed multicolor immunoassay is higher than that of most of the methods. Moreover, the multicolor immunoassay requires no bulky microplate reader, which is suitable for on-site analysis.

### 3.5. Results of Recovery Test

To assess the accuracy of the developed multicolor immunoassay, the recovery test was performed. Firstly, the test of the matrix effect was used to study the anti-interference capacity in real sample analysis. The sample extraction was directly diluted with PBS for multicolor immunoassay, which is simple and convenient. As shown in Figure 8, the matrix effect cloud can be completely removed with a 10-fold or 20-fold dilution for all the samples. Consequently, the optimized dilution for sample analysis was 10-fold due to the lower dilution ratio that achieved higher sensitivity than that of 20-fold. Afterward, the spiked samples were analyzed using a multicolor immunoassay, and the results are summarized in Table 1. The recoveries of reader mode and RGB mode were 81.8–112.4% and 83.3–95.9%, with the coefficient of variable (CVs) of 3.2–16.5% and 2.6–16.4%, respectively. The results of the multicolor immunoassay showed good agreement with the standard HPLC-MS/MS, which can be used for real sample tests.

## 4. Conclusions

This study developed a multicolor immunoassay for the rapid analysis of isocarbophos. The GAM-ALP was used as a signal probe to catalyze AAP into AA, while K_3_[Fe(CN)_6_] and Fe^3+^ were used for the color response of AAP and AA. As the isocarbophos concentration decreased, the AAP-Fe^3+^ complex converted into Prussian blue with a notable color shift from orange to green and then to blue. The absorbance of the AAP-Fe^3+^ complex and Prussian blue was measured to develop a ratio immunoassay for quantitative detection. Moreover, the multicolor conversion can achieve a semiquantitative analysis by the naked eye, which can be used for on-site detection. Furthermore, based on the smartphone and image processing software, the quantitative detection depending on RGB value was developed, which did not require a bulk microplate reader and achieved on-site quantitative analysis. The recovery test indicated the accuracy and practicability of the development of a multicolor immunoassay, which is an ideal tool for the rapid screening of isocarbophos in vegetable and fruit samples. However, some limitations remain in the developed RGB mode analysis. The process of reading RGB signals from microplates still necessitates manual selection of the microplate location within the image, as well as manual fitting of the calibration curve. Furthermore, the field-of-view unmatching between the smartphone camera and the samples resulted in decreased accuracy. Therefore, to enhance accuracy, the following work of RGB analysis will employ the integration of deep learning techniques to reduce the impact of field-of-view unmatching. Moreover, a mobile app will be developed to achieve automatic reading of RGB signals from microplates, fitting of calibration curves, and calculation of results. The developed multicolor immunoassay will be suitable for rapid pesticide screening in low-resource areas, outside laboratories, and in the field so that laboratory testing can be decentralized without bulky lab instruments. This will advance the strengthening of pesticide monitoring by decentralizing on-site analysis.

## Figures and Tables

**Figure 1 foods-13-04057-f001:**
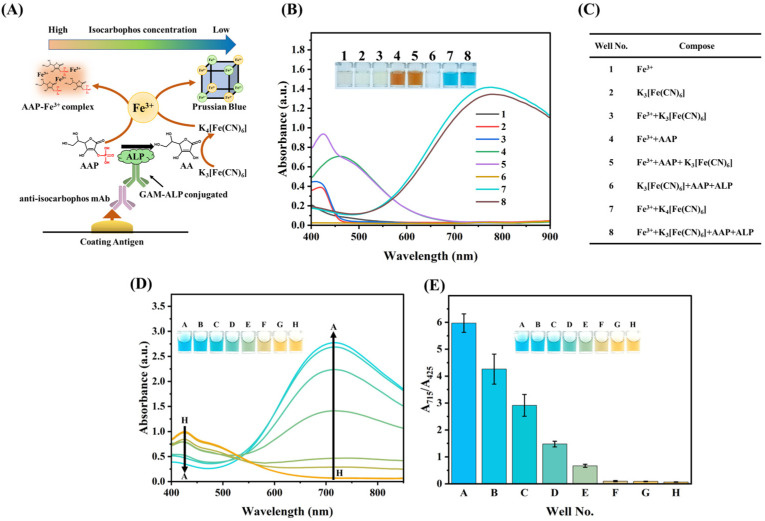
(**A**) The schematic diagram of the multicolor immunoassay; (**B**,**C**) the color reaction of the reagents for the development of multicolor immunoassay; (**D**) the absorption spectrum and (**E**) A_715_/A_425_ ratio for the color conversion of multicolor immunoassay.

**Figure 2 foods-13-04057-f002:**
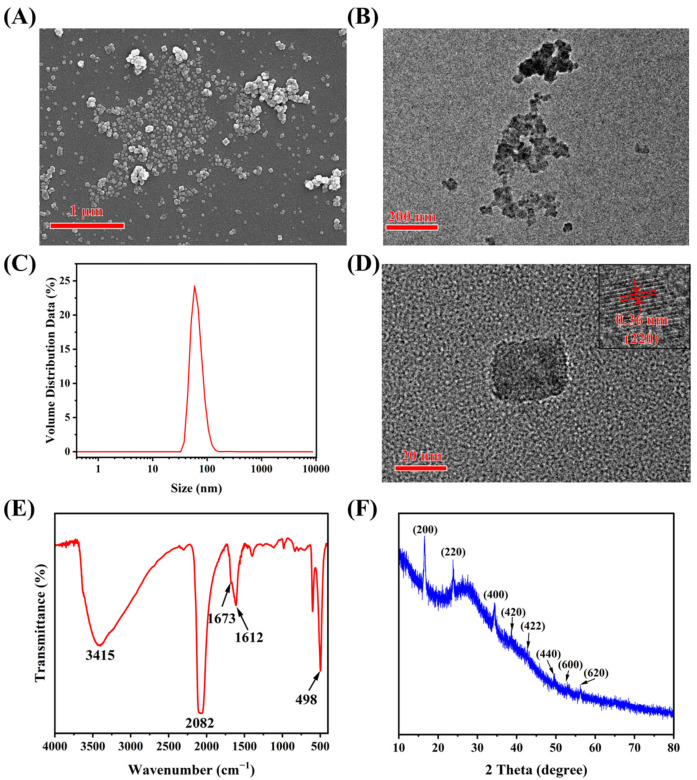
The (**A**) SEM and (**B**) TEM graph of Prussian blue; (**C**) the DLS analysis for Prussian blue; (**D**) the high-resolution image for Prussian blue and the insert graph indicated the lattice fringe of Prussian blue nanocube; the (**E**) FTIR and (**F**) XRD analysis of Prussian blue.

**Figure 3 foods-13-04057-f003:**
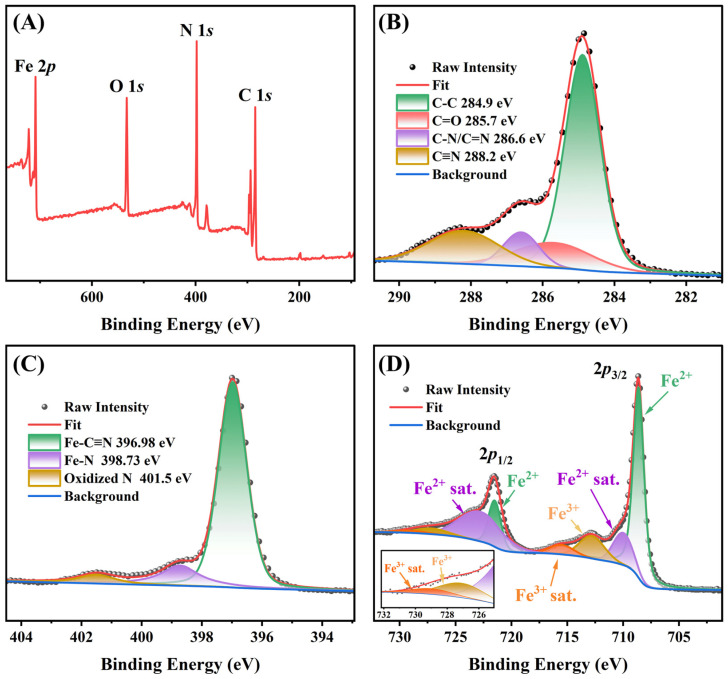
The XPS analysis of Prussian blue; (**A**) survey XPS spectra; (**B**) C 1*s*; (**C**) N 1*s*; (**D**) Fe 2*p*.

**Figure 4 foods-13-04057-f004:**
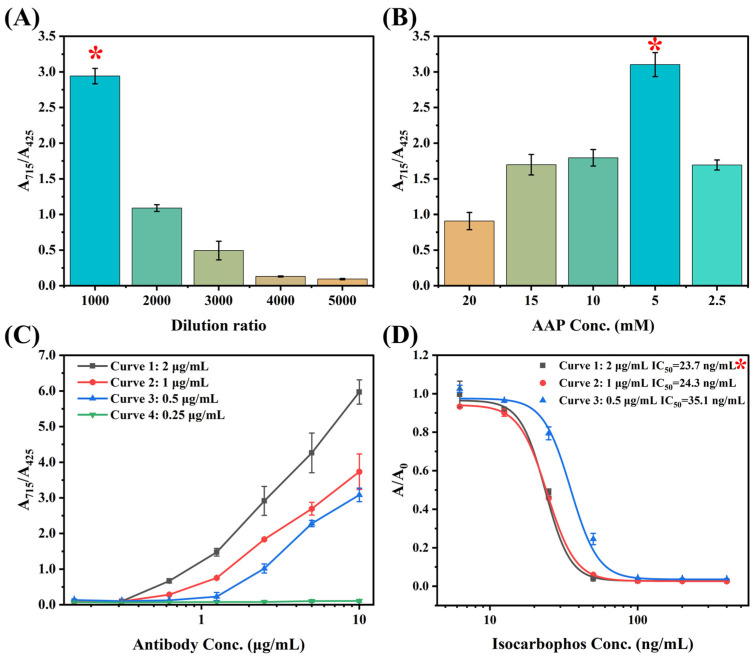
(**A**) The GAM-ALP dilution optimization; (**B**) the AAP concentration optimization; (**C**) the titer for multicolor immunoassay with different concentrations of coating antigen; (**D**) the optimization of the concentration of coating antigen. The red asterisk indicates the optimized parameter.

**Figure 5 foods-13-04057-f005:**
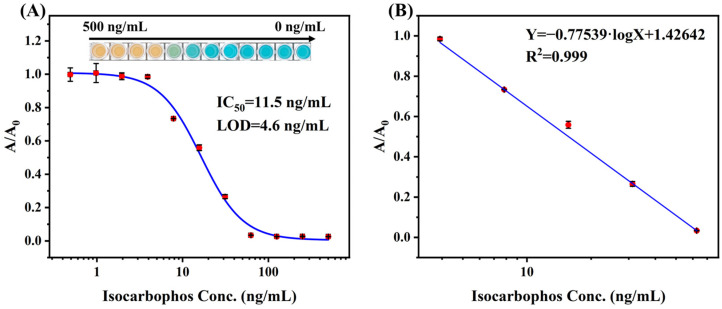
The (**A**) calibration curve and (**B**) linear range of multicolor immunoassay using the microplate reader determination.

**Figure 6 foods-13-04057-f006:**
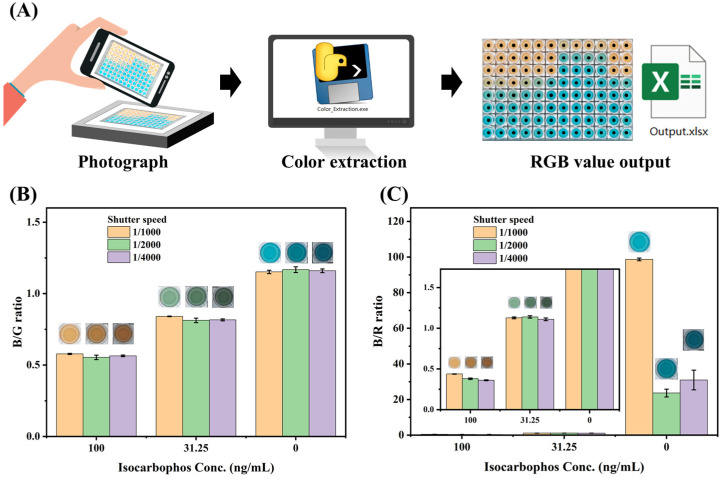
(**A**) The schematic diagram of the RGB mode for the analysis of multicolor immunoassay; (**B**) the B/G mode for the RGB analysis; (**C**) the B/R mode for the RGB analysis.

**Figure 7 foods-13-04057-f007:**
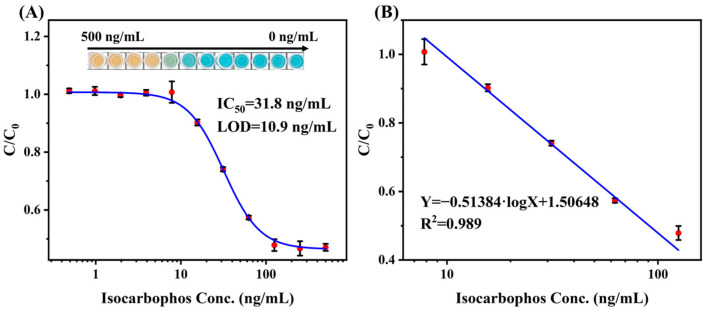
The (**A**) calibration curve and (**B**) linear range of multicolor immunoassay using the RGB value determination.

**Figure 8 foods-13-04057-f008:**
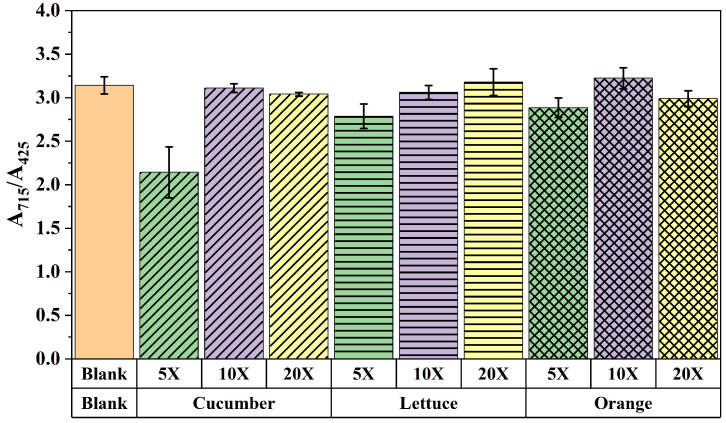
The study of matrix effect with different dilution ratios.

**Table 1 foods-13-04057-t001:** Recoveries of isocarbophos from spiked samples by the multicolor immunoassay and HPLC-MS/MS (n = 3) ***^a^***.

Samples	Spiked (μg/kg)	Multicolor Immunoassay (Microplate Reader Mode)			Multicolor Immunoassay (RGB Analysis Mode)			HPLC-MS/MS		
		Mean (μg/g) (Mean ± SD *^b^*)	Recovery (%)	CV *^c^* (%)	Mean (μg/g) (Mean ± SD)	Recovery (%)	CV (%)	Mean (μg/g) (Mean ± SD)	Recovery (%)	CV (%)
Cucumber	0 *^d^*	ND *^e^*	-	-	ND	-	-	ND	-	-
	2000	1796.3 ± 96.9	89.8	5.4	1780 ± 45.5	89	2.6	1955.2 ± 47.4	97.8	2.4
	1000	818 ± 135	81.8	16.5	894.6 ± 67	89.5	7.5	994.6 ± 25.4	99.5	2.6
	500	422.6 ± 26.2	84.5	6.2	460.4 ± 45.9	92.1	10	466 ± 9	93.2	1.9
Lettuce	0	ND	-	-	ND	-	-	ND	-	-
	2000	2247.8 ± 98.3	112.4	4.4	1665.1 ± 147.1	83.3	8.8	2055.6 ± 110.4	102.8	5.4
	1000	1048.3 ± 58.8	104.8	5.6	864.7 ± 68	86.5	7.9	998.1 ± 70.4	99.8	7.1
	500	470.5 ± 15.2	94.1	3.2	461.4 ± 75.7	92.3	16.4	489.5 ± 32	97.9	6.5
Orange	0	ND	-	-	ND	-	-	ND	-	-
	2000	1831.9 ± 140.1	91.6	7.6	1890.7 ± 274.4	94.5	14.5	1975 ± 112.2	98.8	5.7
	1000	986.6 ± 134.4	98.7	13.6	959.3 ± 67.4	95.9	7	1031.2 ± 59.8	103.1	5.8
	500	462.6 ± 39.4	92.5	8.5	467.5 ± 16.9	93.5	3.6	474.1 ± 23.3	94.8	4.9

***^a^*** For one concentration, three samples were spiked and determined by immunoassay and HPLC-MS/MS; ***^b^*** SD, standard deviation; ***^c^*** CV, coefficient of variance; ***^d^*** blank control; ***^e^*** ND, no data.

## Data Availability

The original contributions presented in the study are included in the article/Appendix A, further inquiries can be directed to the corresponding author.

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
