# Peer review of "Development of a Dual-Readout Multicolor Immunoassay for the Rapid Analysis of Isocarbophos in Vegetable and Fruit Samples"

_foods, 2024, doi:10.3390/foods13244057_

Round 1

Reviewer 1 Report

Comments and Suggestions for Authors

Chen et al., have submitted a detailed research article for the development of multicolor immunoassay for pesticides. I have a few comments on how to improve the current manuscript.

1. The introduction should better emphasize the unique advantages of this multicolor immunoassay and clearly explain how it improves upon or differs from existing methods.

2. Add a brief summary in the "Materials and Methods" section to highlight the key innovations or critical aspects of the methodology for better understanding.

3. Authors can add a detailed table with state-of-the-art techniques for comparison.

4. Authors should emphasize the real-world applications, such as cost-efficiency and accessibility, of using smartphones for RGB analysis to highlight the study's practical relevance.

5. The conclusion summarizes the study well but could include future research opportunities like scaling for commercial use, improving sensitivity, or integrating deep learning and analytical data in detail.

Author Response

Comments 1: The introduction should better emphasize the unique advantages of this multicolor immunoassay and clearly explain how it improves upon or differs from existing methods. Response: Thank you for your comment. The relevant description is provided in introduction (lines 52-63)

Comments 2: Add a brief summary in the "Materials and Methods" section to highlight the key innovations or critical aspects of the methodology for better understanding. Response: Thank you for your comment. The brief summary is provided in Materials and Methods (lines 125-127)

Comments 3: Authors can add a detailed table with state-of-the-art techniques for comparison. Response: Thank you for your comment. The detailed table for comparison is provided in lines 310-315 and in supporting information Table S2.

Comments 4: Authors should emphasize the real-world applications, such as cost-efficiency and accessibility, of using smartphones for RGB analysis to highlight the study's practical relevance. Response: Thank you for your comment. The relevant description is provided (lines 289-295)

Comments 5: The conclusion summarizes the study well but could include future research opportunities like scaling for commercial use, improving sensitivity, or integrating deep learning and analytical data in detail. Response: Thank you for your comment. The relevant description is provided in conclusion (lines 357-368)

Reviewer 2 Report

Comments and Suggestions for Authors

The paper “Development of a Multicolor Immunoassay for the Rapid Analysis of Isocarbophos in Vegetable and Fruit Samples” by Zi-Jian Chen, Wei-Xuan Huang, Hongwu Wang, Mei-Ling Zhang, Kai Chen and Hao Deng describes an innovative strategy in bioanalytics. A competitive immunoassay was designed which uses optical detection via generation of Prussian blue. These phenomena could be followed by naked eye. Furthermore, RGB data can be analysed via a smart phone which leads to quantitative results.   
The system was characterized by SEM, TEM, FT-IR and XRD. Additional, recovery tests for the analytes used were performed. All experimental procedures were thoroughly described and discussed.
Validation by HPLC-MS/MS must be emphasised which confirms the results of rapid analysis procedure.

Author Response

Validation by HPLC-MS/MS must be emphasised which confirms the results of rapid analysis procedure.

Response: Thank you for your comment. The relevant description is provided (lines 329-334)

Reviewer 3 Report

Comments and Suggestions for Authors

This article presents a novel and practical method for detecting isocarbophos residues in fruits and vegetables using a multicolor immunoassay. Its strengths include the innovative integration of RGB-based quantification for real-time analysis and thorough methodological validation. However, broader context, detailed optimization visualization, and language improvements could significantly elevate its impact.

- The abstract: simplify technical jargon; streamlining sentence structure will improve readability for a broader audience. Emphasize the main findings and applications of the multicolor immunoassay at the start will better engage readers.

- the discussion could be expanded to include broader implications, such as comparisons with other rapid detection methods or the potential for commercialization.

- In the optimization section, provide a clear summary table or diagram showing the relationship between variable adjustments (e.g., GAM-ALP dilution, AAP concentration) and assay performance.

- Extend recovery tests to include additional real-world matrices, such as processed food samples, to demonstrate the method's versatility.

- Lines 13–29: Add a concise sentence summarizing the study's broader implications at the end of the abstract.

- Lines 42–45: Add a statement explaining the gap this study addresses in existing pesticide detection methods.

- Lines 98–101: Consider expanding the buffer preparation section with more rationale for buffer selection and its role in the immunoassay.

- Lines 267–270: Include a visual summary (e.g., a table or diagram) showing how adjustments in parameters like GAM-ALP dilution and AAP concentration improved assay performance.

- Line 318: Add a direct comparison with other available detection methods (e.g., ELISA, HPLC-MS/MS).

- Line 335: Add a forward-looking statement about potential challenges or applications.

- Line 176 (Figure 1B): Add clearer annotations to figures showing each step of the color transition for easier understanding.

- Lines 337–339: Expand on how the RGB analysis with smartphones could be further developed, including scalability and potential integration into food safety workflows.

Author Response

- The abstract: simplify technical jargon; streamlining sentence structure will improve readability for a broader audience. Emphasize the main findings and applications of the multicolor immunoassay at the start will better engage readers.

Response:Thank you for your comment. The abstract has been revised according to the reviewer’s comment (lines 14-28).

- the discussion could be expanded to include broader implications, such as comparisons with other rapid detection methods or the potential for commercialization.

Response:Thank you for your comment. The detailed table for comparison is provided in lines 310-315 and in supporting information Table S2.

- In the optimization section, provide a clear summary table or diagram showing the relationship between variable adjustments (e.g., GAM-ALP dilution, AAP concentration) and assay performance.

Response:Thank you for your comment. The results of parameter optimization are summarized in Figure 4 (line 284).

- Extend recovery tests to include additional real-world matrices, such as processed food samples, to demonstrate the method's versatility.

Response:Thank you for your comment. Pesticides mainly residue in raw vegetable or fruit samples. However, agricultural production is scattered, which required on-site rapid detection to for the pesticide residue monitoring (lines 42-48). In this work, raw vegetable or fruit samples were employed in recovery tests to evaluated the accuracy of the multicolor immunoassay. In the future work, other matrix such as animal derived food or processed food samples will be analysis by the developed multicolor immunoassay.

- Lines 13–29: Add a concise sentence summarizing the study's broader implications at the end of the abstract.

Response:The abstract has been revised according to the reviewer’s comment (lines 26-28).

- Lines 42–45: Add a statement explaining the gap this study addresses in existing pesticide detection methods.

Response:According to the reviewer’s comment, relevant statement is added (lines 42-48).

- Lines 98–101: Consider expanding the buffer preparation section with more rationale for buffer selection and its role in the immunoassay.

Response:According to the reviewer’s comment, buffer preparation section is expanded (99-104).

- Lines 267–270: Include a visual summary (e.g., a table or diagram) showing how adjustments in parameters like GAM-ALP dilution and AAP concentration improved assay performance.

Response:The results of parameter optimization are summarized in Figure 4 (line 284)

- Line 318: Add a direct comparison with other available detection methods (e.g., ELISA, HPLC-MS/MS).

Response:Thank you for your comment. The detailed table for comparison is provided in lines 310-315 and in supporting information Table S2.

- Line 335: Add a forward-looking statement about potential challenges or applications.

Response:Thank you for your comment. A forward-looking statement about potential challenges or applications is added (lines 357-368).

- Line 176 (Figure 1B): Add clearer annotations to figures showing each step of the color transition for easier understanding.

Response:Thank you for your comment. Figure 1B is divided into Figure 1B and C for clear description (lines 180-182).

- Lines 337–339: Expand on how the RGB analysis with smartphones could be further developed, including scalability and potential integration into food safety workflows.

Response:Thank you for your comment. The relevant description is provided is added (lines 357-368).

Round 2

Reviewer 3 Report

Comments and Suggestions for Authors

/